# Dynamic Storage Location Assignment in Warehouses Using Deep Reinforcement Learning

**Constantin Waubert de Puiseau** [1,*], **Dimitri Tegomo Nanfack** [2], **Hasan Tercan** [1], **Johannes Löbbert-Plattfaut** [2] **and Tobias Meisen** [1]

1   Institute for Technologies and Management of Digital Transformation, Lise-Meitner-Strasse 27, 42119 Wuppertal, Germany
2   Ingstep GmbH, Ferdinand-Thun-Straße 52B, 42289 Wuppertal, Germany
*   Correspondence: waubert@uni-wuppertal.de

**Abstract:** The warehousing industry is faced with increasing customer demands and growing global competition. A major factor in the efficient operation of warehouses is the strategic storage location assignment of arriving goods, termed the dynamic storage location assignment problem (DSLAP). This paper presents a real-world use case of the DSLAP, in which deep reinforcement learning (DRL) is used to derive a suitable storage location assignment strategy to decrease transportation costs within the warehouse. The DRL agent is trained on historic data of storage and retrieval operations gathered over one year of operation. The evaluation of the agent on new data of two months shows a 6.3% decrease in incurring costs compared to the currently utilized storage location assignment strategy which is based on manual ABC-classifications. Hence, DRL proves to be a competitive solution alternative for the DSLAP and related problems in the warehousing industry.

**Keywords:** warehouse management; logistics; dynamic storage location assignment; reinforcement learning; deep learning; artificial intelligence



## 1. Introduction

The increasing market share of e-commerce and shorter delivery time promises require more flexible and optimized warehouses so that goods are stored and retrieved efficiently. One of the main objectives of operating warehouses lies in the reduction of transport times of pallets from one location to another within the warehouse [1]. The problem of determining where to optimally store goods in a warehouse upon entry or reentry into the system is commonly defined as the Dynamic Storage Location Assignment Problem (DSLAP). The combinatorial nature of the problem as well as uncertainties about the timing of future demands for different goods renders the problem inherently challenging to solve. In other words, whenever a single storage location assignment is needed, the effect of the decision depends on future decisions and required storage operations that may only be predicted to a certain extent a priori. In practice, the storage location assignment task is often handled manually and relies on the expertise of human workers. This expertise predominantly lies in knowledge about the frequency, seasonality, and timing of storage and retrieval operations of goods in warehouses. In the current state of research, more advanced approaches for tackling DSLAP problems exist. They are typically based on a statistical analysis of historical data and warehouse simulations for the derivation of heuristic, metaheuristic, and storage policy-based solution methods [2].

In recent years, machine learning algorithms have increasingly been utilized for the derivation of powerful statistical models in many application domains. For planning problems, deep reinforcement learning (DRL) has emerged as a promising alternative solution approach. It is a machine learning paradigm in which a reinforcement learning agent (RL agent) autonomously derives solution strategies from trial-and-error experiences by updating neural network parameters based on a feedback signal (reward) [3]. DRL is applicable

to sequential decision problems that can be formulated as Markov Decision Processes, i.e., processes in which any next decision can be inferred from the cur-rent situation alone and is independent of previous states of the process [3]. A formal introduction to DRL is presented in Section 4.1. Most famously, DRL has been applied to board and video games, where it resulted in superhuman performance without the supervision and input of experts [4,5]. Driven by these successes, the adaption to use cases in industrial planning problems, such as scheduling problems, has also recently been carried out [6–11]. Since the DSLAP can be formulated as a Markov Decision Process, DRL is theoretically also applicable to the DSLAP and related problems.

In this paper we report a practical case study on a new problem setting in the young field of DRL-based DSLAP solutions. The case study is based on a real-world warehouse from which operational data was stored over the course of 14 months. A DRL agent is trained in simulated re-runs to dynamically assign storage locations with the data of the first twelve months and then evaluated in simulations with data of the last two months. The main contributions of this paper are:

- The empirical proof-of-concept that a real-world DSLAP may be solved end-to-end using DRL.
- Practical design choices for solving the presented DSLAP using DRL.

The remainder of this paper is structured as follows: in Section 2, we discuss related work addressing the DSLAP with machine learning methods. The real-world use case defining the object of study is described in Section 3. Our DRL solution approach is detailed in Section 4. Section 5 covers the experimental setup and used benchmarks. The results are presented in Section 6, followed by a critical discussion of the results in Section 7 and conclusive remarks in Section 8.

## 2. Related Work

The DSLAP has attracted research interest for a long time and many algorithmic solution methods have been proposed. Most recent methods are metaheuristic algorithms [12,13], but tailored solutions based on statistical analysis and manually defined rules [14] or integer linear programming models [15] also continue to be developed. For a survey of solution methods for the DSLAP, we refer the interested reader to Ref. [2].

The capability of machine learning models to derive useful information from warehouse operation data has been investigated in several research works. For example, Li et al. [16] trained a deep learning model to predict the duration-of-stay (DoS), i.e., the time a pallet is going to stay at the assigned location within the warehouse. This prediction is then leveraged in a constraint optimization algorithm to reach the final allocation decision. To obtain a more direct strategy using machine learning, Rimélé et al. [17] proposed a deep learning model to predict the probabilities with which a Monte-Carlo-Tree-Search (MCTS) algorithm would have assigned a particular storage location. To generate a suitably labeled dataset, extensive MCTS runs are required before training. Berns et al. [18], used decision trees to predict zones A, B, or C for pallets entering a simulated warehouse, where each class represents a zone in the warehouse to which the pallet is then transported. All approaches mentioned above have shown performance increases compared to previous methods used in each respective scenario. However, they rely on time-consuming manual labeling by experts or expert systems of historic operations because they are supervised learning methods. In contrast, the method we propose in this case study is based on DRL and therefore does not rely on manual labels.

The general feasibility of DRL for variations of the DSLAP has also been explored recent years. Kim et al. [19] addressed a DSLAP in a ship block stockyard to minimize the rearrangement of ship blocks. In the described setting, ship blocks are assigned a storage location and often must be rearranged due to new unforeseen circumstances. The authors trained two DRL agents, one that assigns the primary location and one that performs all relocations. Another DSLAP variation was studied by Rimélé et al. [20]. Here, a DRL agent was trained to assign pallets to one of six different zones upon new arrival at the warehouse.

The pallet was then transported to the chosen zone by a robot, which picks up a retrieval order on the way back to the entrance. Both publications report substantial improvements of the DRL-based methods over the existing method.

In this paper, we describe a real-world use case of the DLSAP and a methodology to tackle it effectively using DRL. Similarly to the works in Refs. [19,20], we exclusively use DRL for the solution generation, but we address a generally different warehouse logic and layout. In addition, compared to prior publications, no information about future demand is available as a basis for each new storage location assignment. Therefore, the DRL agent, similarly to experts in the warehouse, must learn when and how often certain products are moved solely from historic data and then act accordingly.

## 3. Use Case

In the following we describe the outline, logic and simulation of the studied warehouse, the used data and the implementation details of the reinforcement learning approach.

### 3.1. Warehouse Outline, Logic and Simulation

The object of study is a real-world semi-automatic high-bay warehouse. It has a single point of entry for all arriving and departing goods, where the distribution onto a part of the first rack is carried out manually. The warehouse further consists of twelve lines of high-bay racks which are handled by two automatic storage and retrieval systems (ASTS) which can move freely along guide rails on corridors between racks (see Figure 1). In addition, goods can be moved via conveyor belts between rack 2 and 3, facilitating storage and retrieval from and on racks 3 and 4.

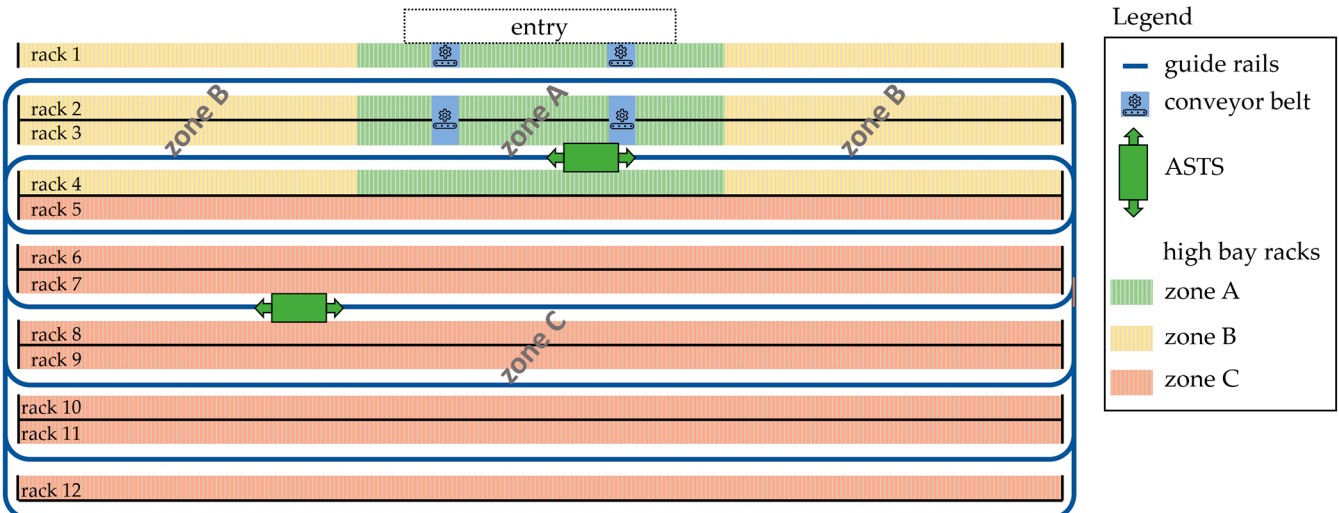

**Figure 1.** Schematic layout of the warehouse.

Operations in the warehouse follow a strict logic: When new goods arrive at the entry point, they are first assigned a storage location in the warehouse. All goods arrive on pallets containing varying amounts of multiple good types. The pallet is manually moved to the first rack and, depending on the assigned storage location, picked up by an ASTS that carries the pallet to the assigned destination. In the afternoon, pallets are often stored in a two-step process: they are first brought to a temporary location close to the entry and then moved to their final location during nighttime. Small conveyor belts installed in racks 1–3 can move pallets between two racks. Usually, one of the ASTS remains in the corridor between racks 3 and 4, because entering this corridor takes a particularly long time. In addition to newly arriving pallets, pallets may be brought to the entry point to retrieve some or all goods on them. If goods remain, a new storage location is determined. Otherwise, the pallet leaves the warehouse. The transport time between storage location assignment and arrival is roughly proportional to the cost of the storage process and therefore constitutes

the objective to be globally minimized. In practice, the storage location assignment takes the form of a manual classification into class A, B or C for each pallet and is based on the experience of the worker. The three classes correspond to zones within the warehouse, for which the transportation time and cost are roughly the same. The zones are distributed as depicted in Figure 1.

The simulation that we used for conducting the experiments represents the described warehouse structure. It stores the current capacity utilization of the racks and keeps track of all transportation costs which arise from conducted storage and retrieval operations. The cost is represented by a unitless metric which corresponds to the mean relative transportation times measured in reality. Correspondingly, transportation operations to zone *A* take one cost unit, to zone *B* two cost units and to zone *C* ten cost units. The capacities of each zone are 9%, 25% and 66% of 9000 total storage locations for zone *A*, *B* and *C*, respectively. As a simplification, we did not consider the two-stage storing process which occurs in the afternoon but treat all pallets as if they were moved to their final location directly. We implemented the simulation logic in Python and follow the OpenAI Gym API guidelines [21], which present a common API for simulations used in reinforcement learning research and practice.

*3.2. Real-World Data*

The used data basis for training and testing of the DRL agent comprises of 12,100 items of historical information on storage and retrieval operations. The data contain all such operations of 500 different goods types collected between January 2021 and April 2022. The information for each operation consists of:

- A timestamp [YYYY-MM-DD hh-mm-ss] when the location assignment took place.
- The loaded good type identification number.
- The number of articles on the pallet.
- The date on which the pallet was first packed and entered the warehouse system.
- The type of storage location assignment (first entry or re-entry after partial retrieval of articles).
- The class (*A*, *B* or *C*), assigned to the pallet based on subjective experience by human workers.

## 4. Reinforcement Learning Approach

In this section, an introduction to DRL and the design choices of the proposed approach are presented regarding the action design of the agent, the observation space, the reward formulation, and the learning algorithm along with its hyperparameters. Unless stated otherwise, they were obtained through preliminary experiments.

*4.1. Introduction to Deep Reinforcement Learning*

Reinforcement learning (RL) can roughly be categorized into value-based and policy-based methods. Value-based methods update *Q*-values representing the expected discounted cumulative reward given the current state *s* for all available actions *a*:

$$Q^{\pi}(s,a) = \mathbb{E}_{\pi}\left[\sum_{k=0}^{\infty} \gamma^k R(s_t|a_t)|s_0 = s,\ a_0 = a\right] \tag{1}$$

The policy of the RL agent during inference is obtained by choosing the action with the maximal corresponding *Q*-value for each state. Since the number of different states in scheduling problems is very large, the function representing the *Q*-values is approximated with a deep neural network and updated using the gathered data during training [3]. A popular representative of value-based methods is DQN [22]. Policy-based methods optimize the policy $\pi$ more directly by learning a function mapping a state s to an action.

This function, in DRL, is also approximated using a deep neural network parameterized with parameters $\theta$. The updates to the policy function follow the gradient

$$\nabla_\theta J(\theta|s) = \mathbb{E}_{\pi_\theta}[Q^\pi(s,a)\cdot\nabla_\theta \log \pi_\theta(s,a)] \tag{2}$$

which was first proposed in Ref. [23]. A popular modern representative is Proximal Policy Optimization (PPO) [24].

### 4.2. Action Space and Interaction with the Simulation

A storage location assignment is required whenever a pallet is in the front of the warehouse (entry) and not empty. The actions the DRL agent may take mirrors the timing and possible choices of the decision-making process that is currently conducted by human workers in the warehouse: the DRL agent may assign a pallet to zone *A*, *B* or *C*. Hence, the action space of the DRL agent is discrete and of size three. The simulation automatically interprets the action and executes the process: changing the location status of the zone and adding the transport cost to the total cost. In the hypothetical situation in which a zone is full, the agent may not choose that zone. This exception is handled by the DRL algorithm and explained in Section 4.5.

### 4.3. Observation Space

In each step, the action should be based on the state of the warehouse, the characteristics of goods on the pallet for which the storage location must be assigned, the storage location assignment and the date. The necessary information on the state of the warehouse is the current capacity utilization in each of the zones. Therefore, the capacity *CAP* of the warehouse is represented as a three-dimensional vector

$$CAP = \left[ \frac{items\ in\ A}{total\ capacity\ of\ A}, \quad \frac{items\ in\ B}{total\ capacity\ of\ B}, \quad \frac{items\ in\ C}{total\ capacity\ of\ C} \right]. \tag{3}$$

The goods information is represented by a vector containing the multi-hot encoded goods type (vector of zeros with length 500). The type of storage location assignment is binary: a 0 indicates that the pallet is entering the warehouse for the first time, whereas a 1 indicates a re-entry. The date is encoded as a single value representing the day of the year. It is scaled through division by 365. From these three features, the agent is supposed to learn, when and how often particular good types are moved in the warehouse. The aggregated resulting observation vector is of length 505.

### 4.4. Reward Design

The chosen reward is directly proportional to the resulting cost of each operation and was scaled such that a stable learning was reached in preliminary experiments. Accordingly, the assignment to zone *A* is rewarded (or rather punished) with $-0.01$, the assignment to zone *B* with $-0.02$ and to zone *C* with $-0.1$ (confer Equation (2)). Note that the sum of all rewards over an observed period will be proportional to the total transportation cost in that time. The agent is supposed to learn to accept temporarily larger punishments for the sake of maintaining enough capacity in less expensive zones for more frequently moved pallets.

$$reward = \begin{cases} -0.01 & if\ action = Zone\ A \\ -0.02 & if\ action = Zone\ B \\ -0.1 & if\ action = Zone\ C \end{cases} \tag{4}$$

### 4.5. Learning Algorithm and Hyperparameters

The chosen DRL-algorithm is PPO [24]. PPO is a popular DRL-algorithm for its stable learning behavior, and it is applicable to the chosen discrete action space design. We compared the performance of PPO with that of DQN [22] in preliminary experiments and observed far superior performance and learning behavior by PPO. Specifically, we

deployed the action-masked version of PPO from the StableBaselines3 implementation [25]. Action masking ensures that no invalid actions, such as assigning a pallet to a full zone, may be taken by the agent during both training and deployment. Although the DRL agent could theoretically learn not to suggest invalid actions through the reward signal, our experiments showed much better learning behavior for the masked version. Suitable hyperparameters were obtained through trial and error. The ones that differ from the defaults of Ref. [25] are listed in Table 1.

**Table 1.** Used hyperparameters for PPO.

| Hyperparameter | Value |
| --- | --- |
| alpha | 0.0001 |
| steps | 19500 |
| gamma | 0.99 |
| ent_coef | 0.00 |
| gae_lambda | 1 |
| vf_coef | 0.5 |
| n_epochs | 10 |
| batch_size | 256 |
| policy_kwargs: net_arch | [256, 256, 256] |

All Code was implemented in Python and executed on an AMD Ryzen 7 4700U (8 MB Cache, 2 GHz) hardware. This relatively limited hardware configuration led to real-world applicable training times of about four hours and inference times for a single storage location allocation decision of 0.7 ms.

## 5. Experimental Setup

### 5.1. Train-Test Split

The division of all historic data into a training dataset and testing dataset is non-trivial, since more training data generally lead to a better generalization to unseen data but more testing data leads to more meaningful evaluations. In this study, the training data includes all data from February 2021 to January 2022. Accordingly, the testing data includes data from February 2022 and March 2022. This way, a seasonality in the frequency of storage and retrieval operations of certain goods, could be learned by the DRL agent from the training data and could be evaluated on the test data.

The training was performed with five different random seeds used for the random initialization of the neural network parameters.

### 5.2. Benchmarks

To evaluate the performance of the DRL agent, four rule-based benchmark storage location assignment methods were implemented:

1. *RANDOM:* The easiest benchmark method samples actions (*A*, *B* or *C*) randomly from a uniform distribution.
2. *Just-in-Order:* This method follows the intuition that the cheapest zones should be used to the limit. Therefore, as long as the capacity utilization of zone A is not 100%, pallets are assigned to zone A. When it is full, pallets are assigned to zone B and so on.
3. *ABC:* This method represents the currently running system in the warehouse. For this benchmark, we use those classes that were assigned by experts and executed in reality.
4. *DoS-Quantiles:* This method is engineered from historic data and serves as the strongest baseline, which can be created only in retrospective. It is based on the duration of stay (DoS) of a certain good type. Two quantiles *q1* and *q2* of the DoS are defined. When the historic average DoS of a good type on a pallet is smaller than or equal to *q1*, the pallet is assigned to zone *A*. If it is between *q1* and *q2*, it is assigned to zone *B*. The rest is assigned to zone C. In a preliminary grid-search of quantile values

$q1 \in [0.35, 0.40, 0.45 \ldots 0.95]$ and $q2 \in [0.40, 0.45, 0.50 \ldots 1.00]$, $q1 = 0.70$ and $q2 = 0.90$ achieved the best results on the whole dataset.

In contrast to all rule-based benchmarks, the DRL agent incorporates knowledge about the current zone capacity utilization, which gives it a theoretical advantage. If one of the benchmark methods assigns a pallet to a full zone, the action is overwritten with the next cheapest zone for an assignment to zones *B* and *C*, and with *B* in case of an invalid assignment to *A*.

## 6. Results

The learning curve of the DRL agent over training steps on the x-axis is depicted in Figure 2. The solid black line indicates the cumulative reward of the DRL agent after each training episode, i.e., after all storage location assignment decisions of one year. It is averaged across the five random seeds. Minimum and maximum values across the random seeds are indicated by the gray shaded area. The horizontal lines depict the cumulative rewards of the benchmarks on the same training data: *DoS-Quantile* (orange), *ABC* (green), *RANDOM* (red) and *Just-in-Order* (blue) from top to bottom in that order. After around 2.5 Mio. executed actions (steps), the DRL agent already performs better than *Just-in-Order* on the training dataset. After 5 Mio. training steps, it consistently beats *RANDOM* and *ABC*. Around 20 Mio. steps into training it converges towards a value between *ABC* and *DoS-Quantile*. Note that, as previously mentioned, *DoS-Quantile* is an artificial benchmark created posteriori that represents an upper limit. It is noteworthy that all five DRL agents train very consistently and differ only marginally in their performance across training.

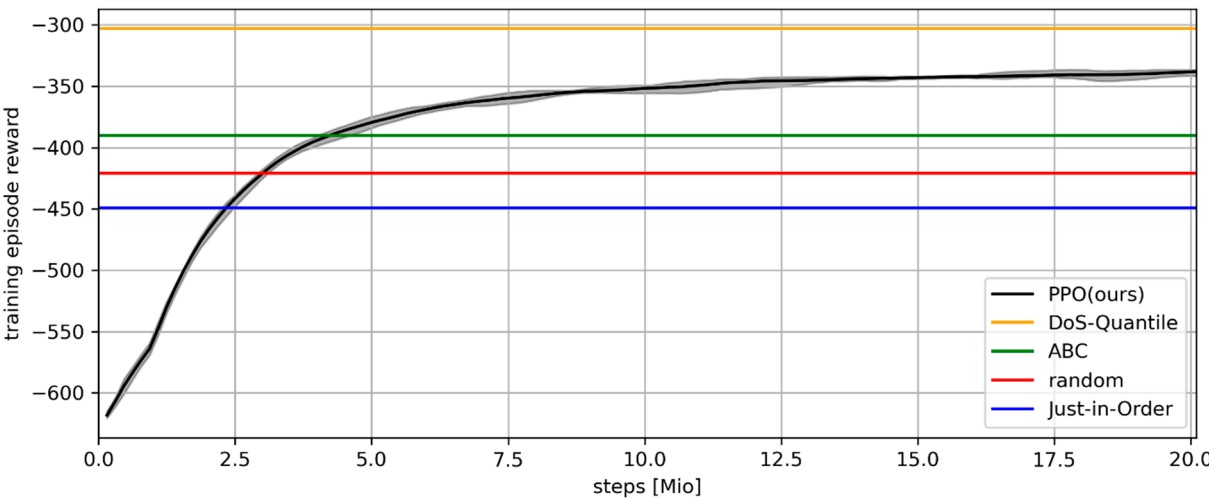

**Figure 2.** Learning curve of the DRL agent displaying the average and standard deviation of the cumulative reward on each training episode over training progress in million steps.

The results on the testing data are summarized in Table 2. The *Total Cost* column lists the unitless cost achieved by the DRL agent (PPO) and all five benchmarks. Thus, lower values represent better performances. The results are qualitatively similarly to the results on the training data: the DRL agent outperforms all realistic benchmark methods, including the reality-based *ABC* method, which it outperforms with 6.3% lower transportation costs.

**Table 2.** Results on the test data. DRL results are boldfaced.

| Agent | Total Cost | Number of Assignments per Zone | | | | Mean DoS per Zone | | |
|---|---|---|---|---|---|---|---|---|
| | | Total | A | B | C | A | B | C |
| **PPO (Ours)** | **37.78** | **1088** | **214** | **647** | **227** | **2.16** | **4.12** | **9.12** |
| DoS-Quantile | 35.99 | 1088 | 257 | 621 | 210 | 2.58 | 3.83 | 16.12 |
| ABC | 40.34 | 1088 | 262 | 561 | 265 | 2.25 | 4.45 | 7.54 |
| RANDOM | 45.43 | 1088 | 369 | 377 | 342 | 4.41 | 4.95 | 4.29 |
| Just-in-Order | 45.70 | 1088 | 110 | 665 | 313 | 4.87 | 3.87 | 6.08 |

The *Number of Assignments per Zone* and *Mean DoS per Zone* columns give insights into the strategy learned by the DRL agent. Since *DoS-Quantile* is the best solution found on the data and *Just-in-Order* the worst, it is helpful to compare the results of PPO to those two. PPO assigns fewer pallets to zone A than *DoS-Quantile*, but more than *Just-in-Order*. One explanation is that PPO is pushed towards the *Just-in-Order* strategy at the beginning of training, because assigning most pallets to zone *A* gives the largest immediate reward. Over the course of training, the agent learns to move away from that short-sighted strategy for the sake of optimizing the cumulative reward across the whole learning episode. Yet, the number of assignments to zone *A* falls below the optimal number, indicating a potential remaining bias towards zone *A*.

The mean DoS per zone indicates that the DRL agent has successfully learned to classify goods into shorter and longer mean DoS in an assigned location. Pallets in zone *A* have the smallest mean DoS when assigned by the DRL agent compared to all benchmarks. However, the best strategy (*DoS-Quantile*) successfully lowers the mean DoS in zone *B*, which seems to be a better strategy, and realizes a very large DoS for zone *C*. This could be an artifact of changing DoS times throughout over time, which can lead to a difference between the training and test data.

## 7. Discussion and Future Work

A reliable reduction of 6.3% in transportation cost is a significant improvement. At the scale of modern warehouses, this brings a substantial competitive advantage. It is worth noting that the presented reinforcement learning approach is easily transferable to warehouses of other industries by modifying only use-case specific details (e.g., goods types and reward signals).

Moreover, we believe that further improvements of the results may be possible but exceed the scope of this case study. In the future we plan to analyze whether the potential bias towards zone *A* mentioned above can be addressed by means of different reward functions. Furthermore, as we have shown in Section 6, the solution strategy found by PPO does not lead to perfect mean DoS for zone C. We expect that the gap between mean DoS times per zone between *DoS-Quantile* and PPO would become narrower with more available training data. This data is constantly gathered in the warehouse and will be used in future studies. Lastly, the results could possibly be further improved through an extended hyperparameter optimization.

Despite the success, there are limitations to the provided methodology, currently still hindering its deployment. The first is a certain difference between the logic implemented in the simulation and reality. In reality, some pallets are preliminarily stored in zone A throughout the day and transported to the other zones at night (compare description of the two-stage storage process in Section 3.1). The slightly different logic may cause overflows in zone A or corrupt the learned strategy of the DRL agent. The second open challenge is posed by arrivals of new goods types, as the observation space of the DRL agent is fixed and depends on the total number of goods types. Therefore, the introduction of a new type of goods makes a re-training necessary. A direction of future research will be the effective handling and re-training for new types of goods.

## 8. Conclusions

This paper presented a successful application of deep reinforcement learning (DRL) to the dynamic storage location assignment problem (DSLAP) using real-world data for training and testing. The trained DRL agent effectively reduces the transportation cost in the warehouse presented in this study by 6.3% compared to the currently used method. The presented approach may easily be transferred to other warehouse layouts and logics. It can therefore be concluded that DRL is a promising approach for DSLAP that should be considered as an assistance system or even automated system when looking for more efficient warehouse operation.

**Author Contributions:** Conceptualization: C.W.d.P., J.L.-P. and T.M.; methodology: C.W.d.P. and D.T.N.; software: D.T.N.; resources: J.L.-P.; writing—original draft preparation: C.W.d.P.; writing—review and editing: H.T. and T.M. All authors have read and agreed to the published version of the manuscript.

**Funding:** This research received no external funding.

**Institutional Review Board Statement:** Not applicable.

**Informed Consent Statement:** Not applicable.

**Data Availability Statement:** Restrictions apply to the availability of these data. Data was obtained from an anonymous company owning the mentioned warehouse and are available upon request from the authors with the permission of said company.

**Conflicts of Interest:** The authors declare no conflict of interest.

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
