# Peer review of "Dynamic Storage Location Assignment in Warehouses Using Deep Reinforcement Learning"

_technologies, doi:10.3390/technologies10060129_

Round 1

Reviewer 1 Report

DSLAP is an important topic in logistics. This paper adopts RL to address this topic, which is a timely research. However, I have the following comments:

1.       The motivation of using RL is not very clear. Why not use there are other methods like metaheuristic or exact algorithms?

2.       The representation of RL is not formal enough. For example, the formulation of rewards should be provided.

3.       There are other RL methods. The authors only use PPO. Why? It would be better to compare other RL methods with PPO.

4.       The literature review is less comprehensive. Some studies on inventory control are missing, for example, A hybrid metaheuristic algorithm for location inventory routing problem with time windows and fuel consumption. Expert Systems With Applications, 2021, 166, 114034

Reviewer 2 Report

What are the challenges and difficulties of the DSLAP problem? I suggest that the authors outline the challenges.

What are the contributions of this paper? Please summarize them.

Is there related work on solving the DSLAP problem using reinforcement learning?Please make it clear.

Section 2: I suggest that the author introduce in this section what is reinforcement learning.

Section 3: What is the purpose of simulating a real repository? Is there any innovation?

 Section 4: How are the states, actions, and rewards of reinforcement learning designed? Please explain the rationale in detail.

Section 5.2: Please cite references to benchmark.

I hope the authors summarize the contribution of the paper and point out the future work in section 8.

Round 2

Reviewer 1 Report

The version is good.

Reviewer 2 Report

The revision is satisfactory, and the revised version could be accepted.